# Developing Monoclonal Antibodies for Immunohistochemistry

**DOI:** 10.3390/cells11020243

**Published:** 2022-01-12

**Authors:** Jacqueline Cordell

**Affiliations:** Nuffield Division of Clinical Laboratory Sciences, Radcliffe Department of Medicine, University of Oxford, Oxford OX3 9DU, UK; jackie.cordell@ndcls.ox.ac.uk

**Keywords:** monoclonal antibodies, immunohistochemistry, synthetic peptides, recombinant proteins

## Abstract

The experiences of a laboratory which pioneered the application of monoclonal antibodies to diagnostic histochemistry is described. This was achieved in four key steps: (1) Monoclonal antibodies were successfully produced to replace the difficult-to-produce and limited polyclonal antibodies available for immunohistochemistry. (2) Monoclonal antibodies were produced to improve the immunoenzymatic detection of bound antibodies, using immunoperoxidase or alkaline phosphatase, increasing sensitivity and allowing the use of two chromogens when applied together. The availability of a reliable alkaline phosphatase-based detection allowed the detection of antigens in tissues with high endogenous peroxidase. (3) Methodologies were developed to unmask antigens not detected in routinely processed paraffin-embedded tissue. (4) Synthetic peptides were used as immunising antigens for the direct production of specific molecules of diagnostic interest. This was expanded to include recombinant proteins. Many reacted with fixed tissue and recognised homologous molecules in other species. In addition to these developments, the laboratory promoted the collaboration and training of researchers to spread the expertise of monoclonal production for diagnosis.

## 1. Introduction

In 1979, I went to work in the Nuffield Department of Pathology, sited at the John Radcliffe Hospital in Oxford. Dr. David Mason had advertised for a research assistant to work with him to produce monoclonal antibodies. David was trained as a haematologist. Within the department there was also Kevin Gatter, a pathologist. Together, they had a vision according to which it would be advantageous if both haematologists and pathologists had some type of histological markers that would help them differentiate one cell type from another. It is often not easy to identify from which cell the tumour or lymphoma originated on the basis of the haematoxylin- and eosin-stained morphology. This was problematic with the increasing use of fiberoptic endoscopy, which yielded specimens that were crushed and too small for an accurate diagnosis. A variety of different non-lymphoid neoplasms may on occasion show a histologic similarity to malignant lymphoma. Histochemical markers specific to leukocyte-associated antigens would make it easier to differentiate between lymphoid and non-lymphoid neoplasms. Their vision was to produce a technique that could be used for diagnosis. A more accurate and reliable diagnosis would allow for a more tailored treatment of patients.

## 2. Monoclonal Antibodies

Polyclonal antibodies had been used to label tissues and cells, and it had been suggested that monoclonal antibodies would prove inferior, in terms of both labelling intensity and specificity, to polyclonal antisera. Notwithstanding this, we decided to go ahead and make our own monoclonal antibodies and see for ourselves.

The production of monoclonal antibodies was first described in 1975 by Köhler and Milstein [1] by a procedure that involves obtaining lymphoid cells from an immunised animal. These cells are immortalised by hybridisation with an established cell line, producing a monoclonal antibody-secreting cell line.

Our first task was to select the immunising antigen. Early efforts used an approach where the antigen was purified, to a greater or lesser degree, from whole cells. These cells were from normal human tissues, for example tonsils, or from blood samples from patients who had neoplastic circulating cells. We also used cultured human cell lines established from various neoplastic diseases. Antigens were primarily selected with a possible diagnostic potential.

These early experiments produced many monoclonal antibodies that were primarily screened on normal and neoplastic tissue cryostat sections and also on patient blood smears. We plated the fusion out into 2 mL 192-well rather than the previously used 200 µL microtitre wells, reducing the number of supernatants to be screened. This method produced many colonies in each well, which we picked out by hand and put into individual 2 mL wells to grow on. Monoclonality was ensured by limiting the dilution. This drastically reduced the loss of positive colonies. Another innovation was to use multiwell slides, which had four sections per slide, separated with screen-printed polytetrofluroethylene so that supernatants were contained in one section.

One of our early obstacles was to persuade the surgeons that if they gave us an extra clinical sample they might gain extra information that might be useful to them in terms of deciding how the treatment of the patient should proceed. It soon became clear that monoclonals were at least as good as polyclonals for immunohistochemistry.

Early antibodies that became invaluable in the diagnosis, such as PD-7/26, a monoclonal antibody against CD45, were described in 1983 when Warnke et al. [2] published a description of two CD45 monoclonal antibodies, PD-7/26 and 2B11. Present on the majority of human leukocytes, CD45 is important in accurately diagnosing lymphoma. Staining with these antibodies indicates, with a very high degree of probability, that a tumour is of white-cell (and usually lymphoid) origin. Furthermore, this monoclonal antibody worked after routine fixation and paraffin embedding, which was not the case for all the monoclonal antibodies that we made. One particular case that I remember, brought to the laboratory by Kevin Gatter, was a particularly crushed sample from a patient with a facial mass. Kevin wanted to know if it was a lymphoma or carcinoma. We were able to tell him that it was a lymphoma. This was of great benefit to the patient, as instead of a brutal facial surgery for a carcinoma they were able to be treated with either chemotherapy or radiation. Some examples of produced reagents are given in Table 1.

## 3. Immunoperoxidase

Of course the usage of monoclonal antibodies for immunohistochemical staining requires good staining methodologies. The lab decided to apply monoclonal antibody technology to Peroxidase:Antiperoxidase (PAP) staining. We made a monoclonal antibody against horseradish peroxidase and used this reagent for the production of mouse PAP. The hybrid cell line P6/38 that we produced enabled PAP complexes to be prepared without any purification steps, simply by adding commercially available horseradish peroxidase to hybridoma supernatants. This gave excellent immunohistological labelling when used in conjunction with a variety of monoclonal antibodies [3]. In general, most surgical biopsy material can be studied satisfactorily with a simple indirect immunoperoxidase or PAP technique. We also used the PAP method to screen for new monoclonal antibody reagents from our fusions, which we could use on these biopsies.

These reagents could be routinely used in our department as a panel of antibodies used for the evaluation of tumours of uncertain origin. These were limited in that they could only be utilised on tissues fixed in acetone such as cryostat sections and blood smears.

## 4. The APAAP Technique

In 1983, we turned our attention towards using immune complexes of alkaline phosphatase monoclonal anti-alkaline phosphatase (APAAP complexes). The APAAP technique gave excellent immunocytochemical labelling reactions comparable in clarity and intensity to those obtained by immunoperoxidase procedures [4]. The APAAP technique, however, had particular value when staining tissues rich in endogenous peroxidase. In 1984, Falini et al. [5] published a paper in which they described the use of APAAP for labelling frozen sections of undecalcified bone marrow biopsies, avoiding the problems of endogenous peroxidase obviating the use of PAP. Subsequently, in 1986, Erber et al. [6] reported in The Lancet the APAAP labelling of blood and bone-marrow samples for phenotypic leukaemia. The APAAP technique was also of immense advantage when it was used in conjunction with immunoperoidase techniques for the double immunoenzymatic staining of pairs of antigens [7].

## 5. Routinely Processed Paraffin-Embedded Sections

The choice of fixative was also an important decision. We wanted to allow the preservation of the antigen in the tissues and make sure that these antigens (which can be easily masked or destroyed) remained visible to the monoclonal antibody. The majority of the early work was performed on Cryostat sections and blood smears, and we quickly came to realise that we needed to be able to use the monoclonal antibodies on routinely fixed paraffin-embedded sections. A considerable effort was given to finding out which antigen retrieval methods could be used to achieve this.

Many methods have been applied to formalin-fixed tissues to try and unmask the antigen so that it is available for a monoclonal antibody to bind. In the beginning, enzymes such as trypsin were used to slightly digest the tissues. Next came the treatment of tissue sections with EDTA solutions. This was moderately successful. The real breakthrough came when tissue sections were put into a solution such as 1 mM EDTA pH 8.0 or pH 6.0 0.1 M sodium citrate and heated in a microwave oven for various lengths of time.

## 6. Anti-Peptide Antibodies

Our other bold approach was to raise antibodies specific to a synthetic peptide that corresponded to an amino acid sequence present in the molecule of interest. Again, we were warned that it would be impossible to make monoclonals against peptides, but we went ahead anyway.

In 1989, Mason et al. [8] published a paper that described using a 12 amino acid peptide sequence from the cytoplasmic domain of the CD3 epsilon chain as an antigen to produce antibodies to CD3, which we hoped would recognise both normal and neoplastic T cells in routinely processed tissue sections. Although five hybridomas were made that detected T cells, the PC3/188 monoclonal antibody was not comparable to polyclonal anti-peptide antibodies in its strength to CD3. It did however show that peptides could be used to create new monoclonal reagents. The purification of anti-sera from rabbits (notably CD3 made with a peptide) on affinity columns produced very pure reagents that could be used in diagnosis.

In collaboration with Albert Tse and the late Alan Williams at The Sir William Dunn School of Pathology in Oxford, we explored the value of using more peptides to raise monoclonal antibodies to specific molecules that could be of interest to the diagnostic community. In 1990, Pezzella et al. [9] published a description of a monoclonal antibody to the bcl-2 oncogene protein. It had previously been reported that bcl-2 expression was specific to human lymphomas in which the 14:18 chromosomal translocation was present. Using synthetic peptides of a 14 amino acid length, we did achieve our aim and made three monoclonals. Using these reagents, we could clearly show that the bcl-2 protein was immunohistochemically detectable in normal lymphoid tissues, that it was absent from proliferating lymphoid cells and that it was expressed in a wide range of neoplastic lymphoproliferative diseases. Thus, the bcl-2 molecule is not specific to the presence of the 14:18 translocation, which refutes earlier work.

Anti-peptide antibodies frequently recognised highly conserved parts of the molecules, and this made them extremely useful to veterinary surgeons and researchers involved in animal research, as they recognized the corresponding molecule in other species. They were able to be used in a menagerie including rats, mice, horses, camels, marsupials, deer and even snakes.

In 1990, JC70A was made, which detected vascular endothelium-associated antigen on routinely processed tissue sections [10]. This was just good fortune. This antibody was raised against a membrane fraction of human hairy cells and recognised a fixation-resistant epitope on endothelial cells in benign and malignant conditions in a wide variety of tissues. Table 2 lists some examples of the produced reagents.

All these antibodies work in formalin-fixed material when using a pre-treatment with either 1 mM EDTA pH 8.0 or pH 6.0 0.1 M sodium citrate.

We also utilised recombinant protein as an immunogen to raise antibodies to specific molecules. One notable example of this is JCB117 (made in collaboration with Marion Brown at The Sir William Dunn School of Pathology in Oxford). This antibody to CD79a stained mantle zone B cells and plasma cells in an identical pattern to previously characterised anti-CD79a antibodies [11]. This antibody is a reliable reagent that can be used by the diagnostic pathologist for the detection of both normal and neoplastic B cells in routinely processed tissue samples.

Both David and Kevin were true collaborators and travelled to many international meetings. Before they left to attend a meeting, they would ask for samples of our best or most recent monoclonal reagents, and these were given to researchers to use in their own laboratories. This led to many productive collaborations and published papers. We had many visiting scientists from around the world who came to work in our laboratories to learn new techniques and take them back to their own institutes. One such collaboration was when Dr. Giovanna Roncador came to Oxford to spend over three years learning how to make and screen monoclonals. She now runs a highly productive group in Madrid that make and utilise their own reagents [12]. Giovanna also set up the very successful European Monoclonal Antibody Network (EuroMAbNet) [12], holding regular international meetings to discuss such matters as antibody validation and protocols with a view to sharing a wealth of expertise in antibody technology.

## 7. Afterword

I would like to emphasise that the key element to our success was the use of tissue sections to screen hybridoma clones. This allowed pan and cross reactions to be eliminated, and when screened by an experienced histologist, interesting reactivities could be pursued. Initially, we relied on serendipity in producing antibodies, but our subsequent efforts were directed towards omissions in our repertoire of diagnostic antibodies. The success of this approach is reflected in the continued extensive use of these reagents. It was a privilege and a joy to work in the Oxford laboratory with David and Kevin, with many lasting relationships being formed with local colleagues and scientists from all over Europe and the world.

## Figures and Tables

**Table 1 cells-11-00243-t001:** Early examples of diagnostic reagents.

Antibody	Target	Usage
TO15	CD22	Frozen tissue only
CR3/43	HLA-DR	Frozen tissue only
CR4/23	Dendritic reticulum cells	Frozen tissue only
TO5	CD35	Pre-treatment for formalin fixed material
F8/86	Factor VIII	Pre-treatment for formalin fixed material
E29	Epithelial membrane antigen	Frozen tissue preferred

**Table 2 cells-11-00243-t002:** Antibodies generated from peptide antigens that become useful diagnostic reagents.

Antibody	Target
HM57	CD79alpha
HM47	CD79alpha
CD5/54	CD5
C8/144	CD8
JC70A	CD31
KP1	CD68
JCB117	CD79alpha

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
