# Peer review of "Developing Monoclonal Antibodies for Immunohistochemistry"

_cells, 2022, doi:10.3390/cells11020243_

Round 1

Reviewer 1 Report

I enjoyed reading this personal perspective on the origins of immunhistochemistry using MoAB on paraffin sections.   

Author Response

Thank you for your kind review

Reviewer 2 Report

This manuscript is a commentary which provides the reader with an insight into the author's academic journey in developing monoclonal antibodies for use in immunohistochemistry (IHC). As it is a personal reflection of the author, there is little to change with respect to the content, however I have a few suggestions to improve the readability and significance:

  • the abstract reads as though the paper is an experimental or methodology paper, yet the commentary does not describe any methods in detail. I would suggest rewriting the abstract to make it clear that this paper is a reflection to avoid readers expecting methodology.
  • Lines 39-42: The author reflects that the initial belief was that monoclonal antibodies would be inferior to monoclonal antibodies for IHC. It would be good to comment on this later in the commentary (perhaps in the Afterword) to indicate whether this was true or proven incorrect once the mabs were isolated.
  • Lines 47-49: It may be of interest to readers to know how particular antigens were selected and how they were purified.
  • Lines 54-59: How was monoclonality ensured? Was there a limiting dilution step amongst the plating/screeing?
  • Lines 75-87: I don't believe this paragraph belongs under the Immunoperoxidase heading.  It could be moved to above the Immunoperoxidiase paragraph.
  • Line 112:  This heading ('Routinely processed paraffin embedded sections') would be better placed before the previous paragraph (at Line 105.

Author Response

The abstract has been revised.

Lines 63-64 notes the efficacy of mono versus polyclonal antibodies.

Lines 51-52 address selection of antigens.

Line 58 covers monoclonality.

I agree that this paragraph should be moved and section 5 heading be moved.

Reviewer 3 Report

This is a fascinating first hand account of how immunohistochemistry with monoclonals has been developed. My only suggestion to change the title of the tables in the manuscript to a more conventional form. For example, Table 1 should be called "Early examples of diagnostic reagents " and should be referred to in the text. The same applies to Table 2.

Author Response

Thank you for your generous review. I have amended the tables as suggested.